# On Computational Limits and Provably Efficient Criteria of Visual Autoregressive Models: A Fine-Grained Complexity Analysis

## Abstract

Recently, Visual Autoregressive (VAR) Models introduced a groundbreaking advancement in the field of image generation, offering a scalable approach through a coarse-to-fine "next-scale prediction" paradigm. Suppose that $n$ represents the height and width of the last VQ code map generated by VAR models, the state-of-the-art algorithm in [Tian, Jiang, Yuan, Peng and Wang, NeurIPS 2024] takes $O(n^{4+o(1)})$ time, which is computationally inefficient. In this work, we analyze the performance bottleneck and hardness result of VAR Models through a fine-grained complexity lens. Our key contribution is identifying the conditions under which VAR computations can achieve sub-quadratic time complexity. We have proved that assuming the Strong Exponential Time Hypothesis (SETH) from fine-grained complexity theory, a sub-quartic time algorithm for VAR models is impossible. To substantiate our theoretical findings, we present efficient constructions leveraging low-rank approximations that align with the derived criteria.

Formally, suppose that $n$ is the height and width of the last VQ code map in VAR models, $d$ is the hidden dimension, $R$ is the bound of the entries of the input matrices for attention calculations in VAR models. We present two results:

- On the negative side, we show that when $d = O(\log n)$ and $R = \Theta(\sqrt{\log n})$, assuming SETH, it is impossible to approximate the output of VAR model up to $1/\text{poly}(n)$ additive error in truly sub-quartic time $O(n^{4-\Omega(1)})$.
- On the positive side, we demonstrate a sharp phase transition in the computational complexity of the VAR model. When $d = O(\log n)$ and $R = o(\sqrt{\log n})$, the runtime improves dramatically from $O(n^{4+o(1)})$ to $O(n^{2+o(1)})$, which approximates the output of the output of VAR model up to $1/\text{poly}(n)$ additive error.

This work initiates the study of the computational efficiency of the VAR model from a theoretical perspective. Our technique will shed light on advancing scalable and efficient image generation in VAR frameworks.

## 1 Introduction

Visual generation technologies now underpin a broad array of applications, ranging from image enhancement (Lin et al., 2025; Guo et al., 2025) and augmented reality (Azad et al., 2024b) to medical diagnostics (Azad et al., 2024a; Ma et al., 2024; Li et al., 2024a) and creative pursuits like game development (Rafner et al., 2020; Chen et al., 2025). By translating text descriptions or other input into detailed and diverse visuals, these models are reshaping both how machines interpret images and how new visual content is created. Leading methods in the field include Variational AutoEncoders (VAE) (Doersch, 2016), Generative Adversarial Networks (GAN) (Goodfellow et al., 2020), and Diffusion models (Ho et al., 2020). Their advancements in producing high-resolution, high-fidelity, and varied imagery have significantly broadened the scope of visual generation, driving improvements in realism, diversity, and overall quality.

The emergence of the Visual AutoRegressive model (VAR) (Tian et al., 2024) marks a notable paradigm shift in image generation. Rather than relying on conventional "next-token prediction",

the VAR model introduces a coarse-to-fine "next-scale prediction" approach, enabling autoregressive transformers to more efficiently learn visual distributions and outperform diffusion-based alternatives. Moreover, the VAR model demonstrates robust zero-shot capabilities in tasks like image inpainting and editing, underscoring its potential for advancing autoregressive models in visual generation.

Despite its demonstrated strengths, there remains a critical need to investigate the VAR model's computational limits and to design efficient algorithms. In (Tian et al., 2024), the authors report that the VAR model has a computational cost of $O(n^4)$, improving upon the $O(n^6)$ complexity associated with earlier autoregressive (AR) methods, where $n$ is the height and width of the last (largest) VQ code map. In this work, we aim to investigate the computational limits and potential efficient algorithms of VAR models. Specifically, we ask the following questions:

*Can we perform the computations of* VAR *models faster than* $O(n^4)$ *time?*

We answer this question affirmatively and summarize our contributions as follows.

- **Computational Limits:** We analyze the computation of the VAR models under the Strong Exponential Time Hypothesis. Let $R$ represent the upper bound of the elements in the input matrices used for attention calculations in VAR models. We establish an upper bound criterion $R^* = \Theta(\sqrt{\log n})$. Crucially, only when $R$ is below this threshold, one can compute VAR models in $O(n^{4-\Omega(1)})$ time (truly sub-quartic time).
- **Provably Efficient Criteria:** We further show that when $R = o(\sqrt{\log n})$, it becomes possible to design an algorithm that approximates the VAR model in almost quadratic time, specifically $O(n^{2+o(1)})$.

## 1.1 OUR RESULTS

Our first result shows that when the attention entry range $R \geq \Omega(\sqrt{\log n})$, it is impossible to design a truly sub-quartic time algorithm. Our results for the lower bound make use of the Strong Exponential Time Hypothesis (SETH) (Impagliazzo & Paturi, 2001) from the area of fine-grained complexity regarding the time required to solve $k$-SAT.

**Theorem 1.1** (Computational Limits of VAR Models, informal version of Theorem 4.5). *Suppose* $d = O(\log n)$ *and* $R = \Theta(\sqrt{\log n})$. *Assuming* SETH*, there is no algorithm that approximates the* VAR *model up to* $1/\operatorname{poly}(n)$ *additive error in* $O(n^{4-\Omega(1)})$ *time.*

Our second result shows that when $R$ is $o(\sqrt{\log n})$, an almost quadratic time algorithm exists:

**Theorem 1.2** (Existence of Almost Quadratic Time Algorithm, informal version of Theorem 5.7). *Suppose* $d = O(\log n)$ *and* $R = o(\sqrt{\log n})$. *There is an algorithm that approximates the* VAR *model up to* $1/\operatorname{poly}(n)$ *additive error in* $O(n^{2+o(1)})$ *time.*

**Roadmap.** Section 2 offers a summary of related work. In Section 3, we outline the mathematical formulation of both the VAR model and its fast version and divide the model into three stages: the VAR Transformer, the feature map reconstruction block, and the VQVAE-Decoder. Section 4 delves into analyzing the computation limits of the VAR model. In Section 5, we examine the running time and error propagation for each block in the fast VAR model and establish the conditions under which the model can be accelerated with proven efficiency. In Section 6, we conclude our contributions.

## 2 RELATED WORK

### 2.1 VISUAL GENERATION MODELS

Recent years have witnessed remarkable advancements in visual generation models, driven by progress in several prominent architectures.

**AutoRegressive Models.** AutoRegressive models for visual generation (Ding et al., 2021; 2022) transform 2D images into 1D token sequences for processing. Early works like PixelCNN (Van den

Oord et al., 2016) and PixelSNAIL (Chen et al., 2018) pioneered pixel-by-pixel generation using a raster-scan approach. Subsequent studies (Razavi et al., 2019; Esser et al., 2021; Lee et al., 2022) extended this concept by generating image tokens in a similar raster order. For example, VQ-GAN (Esser et al., 2021) employs a GPT-2-style decoder-only transformer for image generation, while models such as VQVAE-2 (Razavi et al., 2019) and RQ-Transformer (Lee et al., 2022) enhance this method with additional hierarchical scales or stacked representations. More recently, Visual AutoRegressive (VAR) modeling (Tian et al., 2024) introduced a novel coarse-to-fine "next-scale prediction" approach. This method improves scalability, inference speed, and image quality, outperforming traditional autoregressive techniques and diffusion transformers.

**Diffusion Models.** Diffusion models (Ho et al., 2020; Rombach et al., 2022) are known for their ability to generate high-resolution images by progressively refining noise into coherent visuals. Models such as DiT (Peebles & Xie, 2023) and U-ViT (Bao et al., 2023) exemplify this approach, leveraging probabilistic frameworks to capture underlying data distributions. Recent advancements in diffusion-based generation focus on improving sampling techniques and training efficiency (Song & Ermon, 2019; Song et al., 2020; Lu et al., 2022; Hu et al., 2024a), exploring latent-space learning (Rombach et al., 2022; Wang et al., 2024c;a; Liu et al., 2024), enhancing model architectures (Ho et al., 2022; Peebles & Xie, 2023; Liang et al., 2024c; Wang et al., 2023; Xue et al., 2024), and 3D generation (Poole et al., 2022; Wang et al., 2024b; Xu et al., 2024b).

## 2.2 ACCELERATION VIA LOW-RANK APPROXIMATION

Low-rank approximation has emerged as a powerful technique for addressing the computational challenges associated with modern transformer architectures. By approximating key operations such as attention and gradient computations, these methods significantly reduce the time and resource requirements of training and inference.

**Accelerating Attention Mechanisms.** Due to its quadratic computational complexity with respect to context length, the attention mechanism faces increasing difficulty as the sequence length grows in modern large language models (OpenAI, 2024; AI, 2024; Anthropic, 2024). To tackle this problem, polynomial kernel approximation methods (Aggarwal & Alman, 2022) have been proposed, leveraging low-rank approximations to construct an efficient approximation of the attention matrix. These approaches lead to notable improvements in computation speed, enabling a single attention layer to handle both training and inference tasks with near-linear time complexity (Alman & Song, 2023; 2024c). Additionally, these methods can be extended to more sophisticated attention mechanisms, like tensor attention, while maintaining almost linear time complexity in both training and inference phases (Alman & Song, 2024a). Furthermore, there are works considering RoPE-based attention (Alman & Song, 2024b; Chen et al., 2024b), and differentially private cross attention (Liang et al., 2024f). Additionally, alternative approaches like the conv-basis method introduced in (Liang et al., 2024a) offer further opportunities for accelerating attention computations, providing complementary solutions to this critical bottleneck. Furthermore, there are many other works that use pruning to accelerate attention mechanisms (Liang et al., 2024b; Chen et al., 2024a; Li et al., 2024b; Shen et al., 2025b;a; Hu et al., 2024c; Wu et al., 2024; Xu et al., 2024a).

**Gradient Approximation.** The low-rank approximation is a widely used technique for optimizing transformer training by reducing computational complexity (Liang et al., 2024d;g; Alman & Song, 2024c; Hu et al., 2024b; Chen et al., 2024a; Liang et al., 2024e; Li et al., 2025). Specifically, (Alman & Song, 2024c) builds upon the low-rank approximation framework introduced in (Alman & Song, 2023), which originally focused on forward attention computation, to approximate the gradient of attention mechanisms. This method effectively reduces the computational cost associated with gradient calculations. In (Liang et al., 2024d), this low-rank gradient approximation approach is further extended to multi-layer transformers, demonstrating that backward computations in such architectures can be approximated in nearly linear time. Additionally, (Liang et al., 2024g) generalizes the work of (Alman & Song, 2024c) to a tensor-based attention model, leveraging the forward computation results from (Alman & Song, 2024a) to enable efficient training of tensorized attention mechanisms. Finally, (Hu et al., 2024b) utilizes low-rank approximation methods in the training process of Diffusion Transformers (DiTs)., highlighting the versatility of these methods in various transformer-based models.

# 3 MODEL FORMULATION

Section 3.1 presents the overall architecture of the VAR model and divides its processing workflow into three stages. In Section 3.2, we provide the mathematical formulation for the modules involved in the pyramid-shaped token map generation stage. Section 3.3 offers the mathematical formulation for the modules in the feature map reconstruction stage, while Section 3.4 presents the mathematical formulation for the modules in the VQ-VAE Decoder process stage.

**Notations.** Given an integer $n \in \mathbb{Z}^+$, we use $[n]$ to denote the set $\{1, \ldots, n\}$. Given a vector $c$, the diagonal matrix formed by $c$ is denoted as $\mathrm{diag}(c)$, where $c_i$ denotes the $i$-th diagonal element. Given a matrix $U$, we use $U^\top$ to denote the transpose of $U$. Given a matrix $U$, we denote it Frobenius norm as $\|U\|_F$, which is defined as $\|U\|_F := \sqrt{\sum_{i,j} U_{i,j}^2}$. Additionally, we define $\|U\|_\infty$ as the maximum norm of $U$, which is defined as $\|U\|_\infty = \max_{i,j} |U_{i,j}|$. Given two vectors $c, d \in \mathbb{R}^n$, the notation $c \circ d$ represents the element-wise product (also known as the Hadamard Product). It is defined as: $c \circ d = (c_1 d_1, \ldots, c_n d_n)$. In our paper, nearly linear time is defined as $O(n \operatorname{poly} \log n)$, and almost linear time is defined as $O(n^{1+o(1)})$.

## 3.1 OVERALL ARCHITECTURE

In this section, We present the overall architecture of the VAR model and divide its processing workflow into three stages.

**Stage 1: Pyramid-Shaped Token Maps Generation.** Firstly, the VAR model will start by quantizing an initial input token map $X_{\mathrm{init}} \in \mathbb{R}^{1 \times 1 \times d}$ into $K$ multiple scale pyramid-shaped token maps $(r_1, \ldots, r_K)$, each at an increasingly higher resolution $h_k \times w_k$. During the $k$-th autoregressive step, all the $h_k \times w_k$ will be generated in parallel, conditioned on $r_k$'s prefix $r_1, \ldots, r_{k-1}$. In Section 3.2, we provide a mathematical definition for each module in this stage.

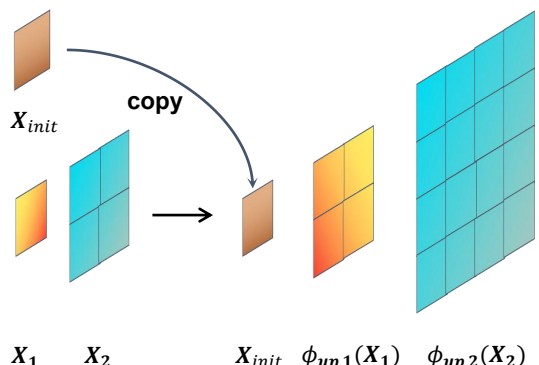

Figure 1: Example of the Pyramid Up-Interpolation Layer $\Phi_{\mathrm{up},2}$ used in the model.

**Stage 2: Feature Map Reconstruction.** The second stage of the VAR model is to reconstruct the generated pyramid-shaped token maps $r_1, \ldots, r_K$ into a Feature Map. Specifically, the VAR model uses an up-interpolation layer to interpolate each of the token maps $(r_1, ..., r_{K-1})$ to the size of $r_K$ and applies a convolution layer to reduce the loss introduced by the interpolation. After this process, the VAR model sums the K token maps to obtain the desired Feature Map. In Section 3.3, we provide a mathematical definition for each module in this stage.

**Stage 3: Generating Image Using VQ-VAE Decoder.** The third stage of VAR model is to use VQ-VAE Decoder to generate the final output image by taking the input of feature map. We follow the implementation of (Tian et al., 2024) and regard the VQ-VAE Decoder as a module composed of fixed-depth ResNet layers, attention layers, and up-interpolation layers. In Section 3.4, we provide a mathematical definition for each module in this stage.

## 3.2 STAGE 1: TOKEN MAPS GENERATION

The VAR model uses the VAR Transformer to convert the initialized token map $X_{\mathrm{init}}$ into a series of pyramid-shaped token maps. The VAR Transformer alternates between up sample blocks and attention layers to get the output.

**Up Sample Blocks.** The $k$-th up sample block takes as input the initial token map $X_{\mathrm{init}}$ and the previous pyramid-shaped token maps $X_1, \ldots, X_k$, sets $Y_1 = X_{\mathrm{init}}$ and up samples each $X_i$ into a new token map $Y_{i+1}$, and outputs the new pyramid-shaped token maps $Y_1, \ldots, Y_{k+1}$.

The upsampling on each token map $X_r (r \in [k])$ uses interpolation with a bicubic spline kernel.

**Definition 3.1** (Bicubic Spline Kernel). *A bicubic spline kernel is a piecewise cubic function $W$ : $\mathbb{R} \to \mathbb{R}$ that satisfies $W(x) \in [0,1]$ for all $x \in \mathbb{R}$.*

**Definition 3.2** (Up-interpolation Layer for One-Step Geometric Sequence). *The layer $\phi_{\text{up},r}$ takes the input feature map $X_r \in \mathbb{R}^{h_r \times w_r \times d}$ and computes the output feature map $Y_{r+1} \in \mathbb{R}^{h_{r+1} \times w_{r+1} \times d}$, where $h_r < h_{r+1}$ are the heights, $w_r < w_{r+1}$ are the widths, and $d \in \mathbb{N}$ is the hidden dimension. It computes $Y_{r+1} = \phi_{\text{up},r}(X_r)$ with a bicubic spline kernel $W$: for $i \in [h_{r+1}], j \in [w_{r+1}], l \in [d]$,*

$$[Y_{r+1}]_{i,j,l} := \sum_{s=-1}^{2} \sum_{t=-1}^{2} W(s) \cdot [X_r]_{\frac{i \cdot h_r}{h_{r+1}}+s, \frac{j \cdot w_r}{w_{r+1}}+t, l} \cdot W(t) \tag{1}$$

We are now ready to present the up sample block $\Phi$.

**Definition 3.3** (Pyramid Up-Interpolation Layer $\Phi$). *The layer $\Phi_{\text{up},k}$ takes the initial token map $X_{\text{init}}$ and the token maps $X_r \in \mathbb{R}^{h_r \times w_r \times c}(r \in [k])$ and computes new token maps $Y_r \in \mathbb{R}^{h_r \times w_r \times c}$. It sets $Y_1 = X_{\text{init}}$ and computes $Y_{r+1} = \phi_{\text{up},r}(X_r)$ as in Definition 3.2. The output is the set consisting $Y_i(i \in [k+1])$.*

**Attention Layer.** After an up sample block, the token maps (after being flattened into a proper shape) will be input into an attention layer.

**Definition 3.4** (Single Attention Layer). *Let $X \in \mathbb{R}^{n \times d}$ denote the input matrix. Let $W_Q, W_K, W_V \in \mathbb{R}^{d \times d}$ denote the weight matrix for query, key, and value, respectively. First, compute the attention matrix $A \in \mathbb{R}^{n \times n}$:*

$$A_{i,j} := \exp(X_{i,*} W_Q W_K^\top X_{j,*}^\top), \ \text{for } i, j \in [n].$$

*Then, compute the output:* $\text{Attn}(X) := D^{-1} A X W_V$, *where* $D := \text{diag}(A\mathbf{1}_n) \in \mathbb{R}^{n \times n}$

**VAR Transformer.** A VAR Transformer with $K$ layers alternates between the attention layer and up sample blocks (where the output of each layer is reshaped to a proper shape as the input for the next layer):

**Definition 3.5** (VAR transformer). *The transformer $\text{TF}$ takes an initial token map $X_{\text{init}} \in \mathbb{R}^{1 \times d}$, computes $Z_0 = X_{\text{init}}$, $Z_k = \Phi_{\text{up},k}(X_{\text{init}}, \text{Attn}_k(Z_{k-1}))$, for $k \in [K-1]$ and finally outputs $\text{Attn}_K(Z_{K-1})$. Here $\Phi_{\text{up},k}$ is defined in Definition 3.3, $\text{Attn}_i$ is defined in Definition 3.4, $Z_{k-1}$ is flatten into shape $(\sum_{r=1}^{k} h_r w_r) \times d$ as input for $\text{Attn}_k$, and the output of $\text{Attn}_k$ is reshaped into $X_r \in \mathbb{R}^{h_r \times w_r \times c}(r \in [k])$ as input for $\Phi_{\text{up},k}$.*

*For convenience, we often abuse notation slightly and write:*

$$\text{TF}(X_{\text{init}}) := \text{Attn}_K \circ \Phi_{\text{up},K-1} \circ \cdots \circ \Phi_{\text{up},1} \circ \text{Attn}_1(X_{\text{init}}),$$

*where $\circ$ denotes function composition.*

### 3.3 Stage 2: Feature Map Reconstruction

In phase 2, the VAR model will transform the generated pyramid-shaped token maps into feature maps. This phase has the following main modules:

**Up Sample Blocks.** The VAR model performs up-sampling on token maps of different sizes, scaling them to the size of the final output feature map. In this process, the VAR model will use the up-interpolation blocks defined in Definition 3.2. To mitigate information loss during token map up-scaling, the VAR model employs convolution blocks to post-process the up-scaled token maps. We define the convolution layers as the following:

**Definition 3.6** (Convolution Layer). *The Convolution Layer is defined as follows:*

- *Let $h \in \mathbb{N}$ denote the height of the input and output feature map.*

- *Let $w \in \mathbb{N}$ denote the width of the input and output feature map.*

- *Let $c_{\text{in}} \in \mathbb{N}$ denote the number of channels of the input feature map.*

- *Let $c_{\text{out}} \in \mathbb{N}$ denote the number of channels of the output feature map.*

- *Let $X \in \mathbb{R}_p^{h \times w \times c_{\text{in}}}$ represent the input feature map.*

- *For $l \in [c_{\text{out}}]$, we use $K^l \in \mathsf{F}_p^{3 \times 3 \times c_{\text{in}}}$ to denote the l-th convolution kernel.*

- *Let $p = 1$ denote the padding of the convolution layer.*

- *Let $s = 1$ denote the stride of the convolution kernel.*

- *Let $Y \in \mathbb{R}_p^{h \times w \times c_{\text{out}}}$ represent the output feature map.*

*We use $\phi_{\text{conv}} : \mathbb{R}_p^{h \times w \times c_{\text{in}}} \to \mathbb{R}_p^{h \times w \times c_{\text{out}}}$ to represent the convolution operation then we have $Y = \phi_{\text{conv}}(X)$. Specifically, for $i \in [h], j \in [w], l \in [c_{\text{out}}]$, we have*

$$Y_{i,j,l} := \sum_{m=1}^{3} \sum_{n=1}^{3} \sum_{c=1}^{c_{\text{in}}} X_{i+m-2, j+n-2, c} \cdot K_{m,n,c}^l + b$$

**Remark 3.7.** *Assumptions of kernel size, padding of the convolution layer, and stride of the convolution kernel are based on the specific implementation of (Tian et al., 2024).*

## 3.4 STAGE 3: VQ-VAE DECODER PROCESS

VAR will use the VQ-VAE Decoder Module to reconstruct the feature map generated in Section 3.3 into a new image. The Decoder of VQ-VAE has the following main modules:

**ResNet Layers.** In the VQVAE decoder, the ResNet block, which includes two (or more) convolution blocks, plays a crucial role in improving the model's ability to reconstruct high-quality outputs. The convolution blocks help capture spatial hierarchies and patterns in the data, while the residual connections facilitate better gradient flow and allow the model to focus on learning the residuals (differences) between the input and output. The definition of convolution block is given in Definition 3.6.

**Attention Layers.** The Attention block helps the Decoder fuse information from different locations during the generation process, which can significantly improve the clarity and detail of the generated images. When applied to a feature map, the attention mechanism computes attention scores for all pairs of pixels, capturing their pairwise relationships and dependencies. The definitions of blocks in attention are given in Section 3.2.

**Up Sample Layers.** The VQ-VAE decoder uses Up-Sample Blocks to progressively increase the spatial resolution of the latent representation. The Up-Sample Blocks in VQVAE combine up-interpolation and convolution blocks to restore the spatial dimensions of the feature maps, facilitating the reconstruction of the high-resolution output image. The convolution block has already been defined in Definition 3.6, and the up-interpolation block has already been defined in Definition 3.2.

## 4 COMPUTATIONAL LIMITS

In this section, we delve into the computational limits of VAR Models, particularly in the context of solving key problems under the assumptions of the Strong Exponential Time Hypothesis (SETH). Section 4.1 introduces SETH as the basis for our complexity analysis. In Section 4.2, we discuss a key result from (Alman & Song, 2023) that establishes the hardness of Approximate Attention Computation. Finally, Section 4.3 presents the lower bound for VAR model efficiency, pinpointing the limitations for sub-quartic performance.

### 4.1 STRONG EXPONENTIAL TIME HYPOTHESIS

We begin by presenting the foundational hypothesis (SETH) (Impagliazzo & Paturi, 2001), which underpins much of our complexity analysis:

**Hypothesis 4.1** (Strong Exponential Time Hypothesis (SETH) (Impagliazzo & Paturi, 2001)). *For every $\epsilon > 0$, there exists a positive integer $k \geq 3$ such that no randomized algorithm can solve k-SAT on formulas with $n$ variables in $O(2^{(1-\epsilon)n})$ time.*

## 4.2 Hardness of Approximate Attention Computation

We begin by introducing the definition of Approximate Attention Computation (AAttC).

**Definition 4.2** (Approximate Attention Computation AAttC$(n, d, B, \delta)$, Definition 1.2 in (Alman & Song, 2023)). *Let $\delta > 0$. Let $R > 0$. Let $X \in \mathbb{R}^{n \times d}$ denote the input of the attention mechanism. Given three matrices $Q, K, V \in \mathbb{R}^{n \times d}$, with the guarantees that $\|Q\|_\infty \leq R$, $\|K\|_\infty \leq R$ and $\|V\|_\infty \leq R$, output a matrix $T \in \mathbb{R}^{n \times d}$ that approximately represents $\mathsf{Attn}(X)$, meaning*

$$\|T - \mathsf{Attn}(X)\|_\infty \leq \delta$$

Now, we are able to give the definition of Fa ast VAR transformer.

**Definition 4.3** (Fast VAR transformer). *Assume the VAR transformer has $m$ attention layers. Let $\Phi_{\mathrm{up},r}$ denote the pyramid up-interpolation layer defined in Definition 3.3. Let $\mathsf{AAttC}_i$ stand for the $i$-th approximate attention computation layer, which is defined in Definition 4.2. Given an input token map $X_{\mathrm{init}} \in \mathbb{R}^{1 \times d}$. Let $l = \sum_{i=1}^{K} h_i w_i$. We define the fast VAR transformer as the following*

$$\mathsf{FTF}(X_{\mathrm{init}}) := \mathsf{AAttC}_K \circ \Phi_{\mathrm{up},K-1} \circ \cdots \circ \mathsf{AAttC}_2 \circ \Phi_{\mathrm{up},1} \circ \mathsf{AAttC}_1(X_{\mathrm{init}}) \in \mathbb{R}^{l \times d},$$

*In this expression, $\circ$ stands for functional composition.*

Next, we state a result for the Approximate Attention Computation (AAttC) from (Alman & Song, 2023).

**Lemma 4.4** (Theorem 4.6 in (Alman & Song, 2023)). *Suppose $d = O(\log n)$ and $R = \Theta(\sqrt{\log n})$. Assuming SETH, there is no algorithm that solves the Approximate Attention Computation (AAttC) up to $1/\operatorname{poly}(n)$ additive error in $O(n^{4-\Omega(1)})$ time.*

## 4.3 Computational Limits of Fast VAR Models

We now present a main theorem detailing the lower bound for VAR model computation.

**Theorem 4.5** (Computational Limits of Fast VAR Models, formal version of Theorem 1.1). *Suppose $d = O(\log n)$ and $R = \Theta(\sqrt{\log n})$. Assuming SETH, there is no algorithm that approximates the VAR model up to $1/\operatorname{poly}(n)$ additive error in $O(n^{4-\Omega(1)})$ time.*

*Proof.* By Lemma 4.4, in the $K$-th step ($K = \log_a n$), the VAR Transformer must compute attention with a computational cost at least

$$\Omega(L_K^{2-q} \cdot d) = \Omega((\sum_{i=1}^{K} (\alpha^{i-1})^2)^{2-q} \cdot d) \geq \Omega((\alpha^{2K} - 1)^{2-q} \cdot d) \geq \Omega(n^{4-2q} d).$$

In the first step above, we use the definition of $L_K$. The second step applies the standard geometric series formula, and the final inequality is due to the fact $K = \log_a n$. $\qquad\square$

In Theorem 4.5, we show our hardness result. The theorem states that we can't accurately approximate ($\epsilon = 1/\operatorname{poly}(n)$) the output of VAR model in $O(n^{4-\Omega(1)})$ time when SETH holds. This implies that, under the condition of SETH, achieving an efficient approximation algorithm for the VAR model within this time bound is infeasible. As a result, we set a lower bound on the complexity of approximating the output of the VAR model, indicating that any algorithm aimed at approximating the model with a small error will need to operate in at least $O(n^{4-\Omega(1)})$ time in the worst case.

## 5 Provably Efficient Criteria

Section 5.1 details the running time of the fast VAR Transformer, feature map reconstruction layer, and Fast VQ-VAE Decoder. In Section 5.2, we analyze the error propagation in both the Fast VAR Transformer and the Fast VQ-VAE Decoder. Section 5.3 presents our findings regarding the existence of an almost quadratic time algorithm.

## 5.1 RUNNING TIME

Here, we present an overview of the computational cost associated with the Fast VAR Transformer, feature map reconstruction block, and Fast VQ-VAE Decoder.

Firstly, we show that the runtime of the VAR Transformer can be sped up to $O(n^{2+o(1)})$.

**Lemma 5.1** (Running time of Fast VAR Transformer, informal version of Lemma E.3). *Assume the fast* VAR *transformer defined in Definition 4.3 has $K$ attention layers. Let $k_1 \in [K]$. Let $X_{\mathrm{init}} \in \mathbb{R}^{1 \times 1 \times d}$ denote the first scale token map. Let $\alpha > 1$ denote the growth rate of the height and width of the token map at each level. Then for $k_1 \in [K]$, the $k_1$-th token map $r_{k_1} \in \mathbb{R}^{\alpha^{k_1-1} \times \alpha^{k_1-1} \times d}$. Let $r_K \in \mathbb{R}^{n \times n \times d}$ denote the last scale token map, where $n = \alpha^{K-1}$. Let $d = O(\log n)$ denote the embedding size of each token.*

*Then, the total runtime of the* VAR *Transformer for generating token maps can be accelerated to $O(n^{2+o(1)})$.*

*Proof.* Please refer to Lemma E.3 for the proof's details. $\square$

Then, we proceed to show that the runtime of the feature map reconstruction layer is $O(n^{2+o(1)})$.

**Lemma 5.2** (Running time of Feature Map Reconstruction Layer, informal version of Lemma E.4). *Let $X_{\mathrm{init}} \in \mathbb{R}^{1 \times 1 \times d}$ denote the initial token map. Let $\alpha > 1$ denote the growth rate of the height and width of the token map at each level. Then for $k \in [K]$, the $k$-th token map $r_k \in \mathbb{R}^{\alpha^{k-1} \times \alpha^{k-1} \times d}$. Let $r_K \in \mathbb{R}^{n \times n \times d}$ denote the last scale token map, where $n = \alpha^{K-1}$. Let $d = O(\log n)$ denote the embedding size of each token.*

*Then, the total runtime of the Feature Map Reconstruction Layer is $O(n^{2+o(1)})$.*

*Proof.* Please refer to Lemma E.4 for the proof's details. $\square$

Finally, we show that the runtime of the VQVAE Decoder can be sped up to $O(n^{2+o(1)})$.

**Lemma 5.3** (Running time of Fast VQ-VAE Decoder, informal version of Lemma E.6). *Let $k_1, k_2, k_3 \in \mathbb{N}$ be constant numbers. Given $X \in \mathbb{R}^{n \times n \times d}$ as the input feature map. Let $d = O(\log n)$. Assume that there are $k_1$ up-interpolation layers $\phi_{\mathrm{up}}$ defined in Definition 3.2. Given a feature map $M \in \mathbb{R}^{h \times w \times d}$. For $i \in [k_1]$, we assume $i$-th up-interpolation layer's output $\phi_{\mathrm{up}}^i(M) \in \mathbb{R}^{O(h) \times O(w) \times d}$. We assume there are $k_2$ approximate attention layers $\mathsf{AAttC}$ defined in Definition 4.2. Given a feature map $M \in \mathbb{R}^{h \times w \times d}$. For $i \in [k_1]$, the $i$-th approximate attention layer's output $\mathsf{AAttC}(M) \in \mathbb{R}^{h \times w \times d}$. We assume there are $k_3$ convolution layers $\phi_{\mathrm{conv}}$ defined in Definition 3.6. Given a feature map $M \in \mathbb{R}^{h \times w \times d}$. For $i \in [k_1]$, we assume $i$-th convolution layer's output $\phi_{\mathrm{conv}}^i(M) \in \mathbb{R}^{h \times w \times O(d)}$.*

*Then, the total runtime of the VQ-VAE decoder can be accelerated to $O(n^{2+o(1)})$.*

*Proof.* Please refer to Lemma E.6 for the proof's details. $\square$

## 5.2 ERROR PROPAGATION ANALYSIS

In this section, we present an error analysis introduced by the fast algorithm applied to the VAR and VQVAE models.

Firstly, we can show that the model output error introduced by the fast algorithm for VAR will not exceed $1/\mathrm{poly}(n)$.

**Lemma 5.4** (Error analysis for the Fast VAR Transformer, informal version of Lemma C.7). *Let $X_{\mathrm{init}} \in \mathbb{R}^{1 \times d}$ denote the initial input token map. Let $K \in \mathbb{N}$ represent the number of approximate attention layers in the* VAR *model. Let the* VAR *transformer $\mathsf{TF}_K$ be defined as Definition 3.5. Let the Fast* VAR *Transformer $\mathsf{FTF}_K$ as given in Definition 4.3. For $i \in [K]$, let $\mathsf{TF}_i(X_{\mathrm{init}})$ denote the output of the $i$-th iteration of the* VAR *transformer. For $i \in [K]$, let $\mathsf{FTF}_i(X_{\mathrm{init}})$ denote the output of the $i$-th iteration of the fast* VAR *transformer. Let $\mathsf{TF}_K(X_{\mathrm{init}}) \in \mathbb{R}^{O(n^2) \times d}$ denote the*

*final output of the* VAR *transformer. Let* $\mathsf{FTF}_K(X_{\mathrm{init}}) \in \mathbb{R}^{O(n^2) \times d}$ *denote the final output of the fast* VAR *transformer. Assume each entry in the matrices can be represented using* $O(\log n)$ *bits. Let* $U, V \in \mathbb{R}^{n \times k}$ *be low-rank matrices constructed for polynomial approximation of attention matrix* $\mathsf{AAttC}(X)$. *Let* $f$ *be a polynomial with degree* $g$. *Then, we can show that the error bound of the final output* $\mathsf{FTF}_K(X_{\mathrm{init}})$ *as* $\|\mathsf{FTF}_K(X_{\mathrm{init}}) - \mathsf{TF}_K(X_{\mathrm{init}})\|_\infty \le 1/\operatorname{poly}(n)$

**Remark 5.5.** *Since the modules in the Feature Map Reconstruction stage (Stage 2) only consist of an up-interpolation layer and a convolution layer, without any attention layers, the acceleration method proposed in this paper cannot be applied to this stage. Moreover, since the Feature Map Reconstruction phase only involves simple linear operations, it is trivial that this stage will not introduce more than* $1/\operatorname{poly}(n)$ *error.*

We also present that the model output error introduced by the fast algorithm for the VQ-VAE Decoder will not exceed $1/\operatorname{poly}(n)$.

**Lemma 5.6** (Error analysis of Fast VQ-VAE Decoder, informal version of Lemma D.2). *Let* $X \in \mathbb{R}^{n \times d}$ *denote the input matrix. Let the up-interpolation Layer* $\phi_{\mathrm{up}}$ *be defined as Definition 3.2. Let the convolution layer* $\phi_{\mathrm{conv}}$ *be defined as Definition 3.6. Let the attention layer* $\mathsf{Attn}$ *be defined as Definition 3.4. Let the fast attention layer* $\mathsf{AAttC}$ *be defined as Definition 4.2. Let the VQ-VAE Decoder be the composition of a constant number of up-interpolation layers, convolutions layers, and attention layers. Let the Fast VQ-VAE Decoder be defined as substituting all* $\mathsf{Attn}$ *layers in VQ-VAE with* $\mathsf{AAttC}$ *layers. Then, we can show that the approximation error of the Fast VQ-VAE Decoder can be bounded by* $1/\operatorname{poly}(n)$.

### 5.3 Existence of Almost Quadratic Time Algorithm

This section presents a theorem proving the existence of a quadratic-time algorithm that speeds up the VAR model and guarantees a bounded additive error.

**Theorem 5.7** (Existence of Almost Quadratic Time Algorithm, formal version of Theorem 1.2). *Suppose* $d = O(\log n)$ *and* $R = o(\sqrt{\log n})$. *There is an algorithm that approximates the* VAR *model up to* $1/\operatorname{poly}(n)$ *additive error in* $O(n^{2+o(1)})$ *time.*

*Proof.* By combining the result of Lemma 5.1, Lemma 5.2, Lemma 5.3, Lemma 5.4 and Lemma 5.6, we can easily derive the proof. $\square$

In Theorem 5.7, we show that we can accurately approximates ($\epsilon = 1/\operatorname{poly}(n)$) the overall VAR model output in almost quadratic time $n^{2+o(1)}$ under practical assumptions. By Lemma 5.1, Lemma 5.2 and Lemma 5.3, we know that the bottleneck in accelerating the runtime of the VAR model is the attention computation (The origin running time cost is $O(n^{4+o(1)})$). The insight of the theorem is that we apply approximate attention computation $\mathsf{AAttC}$ (The running time cost is $O(n^{2+o(1)})$) to replace the original attention computation $\mathsf{Attn}$. In this way, our methods solve the VAR model acceleration. This result enables the VAR model to significantly accelerate the inference process in image generation, making it more competitive in the field of image generation. We provide more discussion in Appendix A.

## 6 Conclusion

This paper provides a fine-grained complexity analysis of Visual Autoregressive (VAR) models, identifying computational limits and efficient criteria under the Strong Exponential Time Hypothesis (SETH). By rigorously analyzing computational trade-offs and proposing provably efficient criteria, this work establishes a foundational understanding that will guide the development of next-generation autoregressive models in visual generation. We demonstrate the infeasibility of achieving sub-quartic time complexity for VAR computations when the norm of input matrices exceeds a critical threshold. In contrast, we establish that sub-quadratic time approximations become feasible under carefully designed conditions, leveraging low-rank approximations. In future works, we will explore the extension of these methods to other domains where autoregressive models play a pivotal role, such as text-to-image synthesis and multi-modal generation tasks. Additionally, integrating hardware acceleration strategies could further optimize the computational pipeline, broadening the applicability of VAR models in resource-constrained environments.

## ETHIC STATEMENT

This paper does not involve human subjects, personally identifiable data, or sensitive applications. We do not foresee direct ethical risks. We follow the ICLR Code of Ethics and affirm that all aspects of this research comply with the principles of fairness, transparency, and integrity.

## REPRODUCIBILITY STATEMENT

We ensure reproducibility of our theoretical results by including all formal assumptions, definitions, and complete proofs in the appendix. The main text states each theorem clearly and refers to the detailed proofs. No external data or software is required.

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

# Appendix

**Roadmap.** In Section A, we discuss the potential impacts and future directions. Section B introduces key notations. Section C details the error analysis for the VAR Transformer. In Section D, we present the error analysis of the VQVAE decoder. In Section E, we evaluate the running time of VAR models and fast VAR models.

## A    DISCUSSION

The fine-grained analysis of Visual Autoregressive (VAR) models we provided in this paper uncovers the critical computational limitations and proposes criteria that ensure efficiency under the Strong Exponential Time Hypothesis (SETH). The insights from this analysis are not only important for VAR models but also carry broader implications for the deep learning and machine learning communities as a whole. One of the key contributions of this work is that understanding the computational bottlenecks of VAR models allows us to more clearly delineate the theoretical boundaries of model performance, which in turn helps guide the design of future models.

By exploring the specific conditions under which the VAR models hit computational limits, it is important to identify and address these bottlenecks early in the model development process. This understanding can prevent the misallocation of resources toward achieving computational feats that are not feasible, particularly in the context of autoregressive models used for visual generation tasks. In particular, demonstrating that sub-quartic time complexity is unattainable when input matrices exceed a critical threshold provides a crucial reference point for the deep learning community. This knowledge empowers researchers to set realistic expectations regarding model efficiency and to focus their efforts on optimizations that are computationally viable.

This work provides a foundational framework for understanding and overcoming the computational bottlenecks in generative models. It will serve as a key resource for researchers striving to design the next generation of efficient autoregressive models. By addressing the limitations of current models and offering clear guidance on how to optimize them, we hope to inspire more efficient and scalable solutions for a wide array of machine-learning applications, extending far beyond visual generation.

## B    NOTATIONS

Given an integer $n \in \mathbb{Z}^+ \cup \{0\}$, the set $\{1, 2, \ldots, n\}$ is represented by $[n]$. In our paper, nearly linear time is defined as $O(n \operatorname{poly} \log n)$, and almost linear time is defined as $O(n^{1+o(1)})$. Given a vector $c$, the diagonal matrix formed from $c$ is denoted as $\operatorname{diag}(c)$, where $c_i$ is the $i$-th diagonal entry of this matrix. Given a matrix $U$, we use $U^\top$ to denote the transpose of $U$. Given two vectors $a$ and $b$, which have the same length. The element-wise multiplication of $c$ and $d$ is denoted as $c \circ d$ with $i$-th entry being $c_i d_i$. Given a matrix $U$, we use $\|U\|_F$ to represent the Frobenius norm of $U$. Specifically, we have $\|U\|_F := \sqrt{\sum_{i,j} U_{i,j}^2}$. Given a matrix $U$, we use $\|U\|_\infty$ to represent the maximum norm of $U$. Specifically, we have $\|U\|_\infty := \max_{i,j} |U_{i,j}|$.

## C    ERROR ANALYSIS OF FAST VISUAL AUTO-REGRESSIVE TRANSFORMER

This section focuses on the error analysis of the VAR model. In Section C.1, we introduce the Lipschitz property of a polynomial function. In Section C.2, we analyze the error propagation of inner product operation by giving two approximated inputs. In Section C.3, we analyze the error propagation of the fast attention $\mathsf{AAttC}(X)$. In Section C.4, we analyze the error between $\mathsf{AAttC}(X')$ and $\mathsf{Attn}(X)$ where $X'$ is the approximated value of $X$. In Section C.5, we conduct an error analysis of the up-interpolation layer. In Section C.6, we conduct the error analysis of the VAR transformer. Finally, in Section C.7, we conduct the error analysis of the Fast VAR transformer.

### C.1    LIPSCHITZ OF POLYNOMIAL

In this section, we introduce the Lipschitz property of polynomials.

**Lemma C.1** (Lipschitz of polynomial). *Assuming the conditions below are satisfied:*

- *Let $x \in \mathbb{R}$.*

- *Let $x' \in \mathbb{R}$ denote the approximated version of $x$.*

- *Let $R > 1$.*

- *Suppose we have $|x| \leq R, |x'| \leq R$.*

- *Let $f$ be a polynomial with degree $g$.*

*We can then demonstrate that:*

$$|f(x) - f(x')| \leq O(R^{g-1}) \cdot |x - x'| \tag{2}$$

*Proof.* Firstly, we can show

$$f(x) = a_g x^g + \cdots + a_1 x + a_0$$

where for each $i \in [g]$, $a_i \in \mathbb{R}$.

Thus, we can show

$$
\begin{aligned}
|f(x) - f(x')| &= |\sum_{i=1}^{g} a_i(x^i - x'^i)| \\
&\leq \sum_{i=1}^{g} |a_i(x^i - x'^i)| \\
&= \sum_{i=1}^{g} |a_i \cdot (x - x') \cdot \sum_{j=0}^{i-1} x^j x'^{i-1-j}|
\end{aligned}
$$

The first step is derived from Eq. (2), the second from the triangle inequality, and the final step from simple algebra.

Then, we can move forward to show that

$$
\begin{aligned}
&\sum_{i=1}^{g} |a_i \cdot (x - x') \cdot \sum_{j=0}^{i-1} x^j x'^{i-1-j}| \\
&\leq \sum_{i=1}^{g} |a_i \cdot (x - x')| \cdot i \cdot R^{i-1} \\
&= |x - x'| \cdot \sum_{i=1}^{g} |a_i| \cdot i \cdot R^{i-1} \\
&\leq O(R^{g-1}) \cdot |x - x'|
\end{aligned}
$$

The first step above is a consequence of the condition that $|x| \leq R$ and $|x'| \leq R$, And the second and last step derive from basic algebra. $\square$

## C.2 ERROR PROPAGATION OF INNER PRODUCT

In this section, we conduct the error analysis of the inner product operation given two approximated inputs $u'$ and $v'$.

**Lemma C.2** (Error propagation of inner product). *Assuming the conditions below are satisfied:*

- *Let $u, v \in \mathbb{R}^k$ denote two vectors.*

- *Define $u', v' \in \mathbb{R}^k$ as the approximation value of $u, v$.*

- *Let $R > 1$.*

- *Assume the value of each entry in matrices can be bounded by $R$.*

- *Let $\epsilon \in (0, 0.1)$ denote the initial approximation error.*

- *Suppose we have $\max\{\|u' - u\|_\infty, \|v' - v\|_\infty\} \leq \epsilon$.*

*Then, we can prove that*

$$|\langle u', v'\rangle - \langle u, v\rangle| \leq 2k\epsilon R$$

*Proof.* Firstly, we can show that

$$|\langle u', v'\rangle - \langle u, v\rangle| = |\sum_{i=1}^{k}(u_1' v_1' - u_1 v_1)|$$

$$\leq \sum_{i=1}^{k}|u_1' v_1' - u_1 v_1|$$

$$\leq \sum_{i=1}^{k}|u_1'(v_1' - v_1) + v_1(u_1' - u_1)|$$

$$\leq \sum_{i=1}^{k}(|u_1'| \cdot |v_1' - v_1| + |v_1| \cdot |u_1' - u_1|)$$

The first step results from simple algebra, the second from triangle inequality, the third from algebraic manipulation, and the final step again from triangle inequality.

Then, we can move forward to show

$$\sum_{i=1}^{k}(|u_1'| \cdot |v_1' - v_1| + |v_1| \cdot |u_1' - u_1|)$$

$$\leq \sum_{i=1}^{k} 2 \cdot R \cdot \epsilon$$

$$\leq 2k\epsilon R$$

The first step is derived from the conditions that each entry is at most $R$, $\|u' - u\|_\infty \leq \epsilon$ and $\|v' - v\|_\infty \leq \epsilon$. The second step follows directly from algebraic manipulation.

$\square$

### C.3    ERROR ANALYSIS OF $\mathsf{AAttC}(X')$ AND $\mathsf{AAttC}(X)$

This section presents the error analysis between $\mathsf{AAttC}(X')$ and $\mathsf{AAttC}(X)$.

**Lemma C.3** (Error analysis of $\mathsf{AAttC}(X')$ and $\mathsf{AAttC}(X)$)**.** *Assuming the conditions below are satisfied:*

- *Let $X \in \mathbb{R}^{n \times d}$ denote the input matrix.*

- *Define $X' \in \mathbb{R}^{n \times d}$ as the approximation version of input matrix.*

- *Let $\epsilon \in (0, 0.1)$ denote the approximation error.*

- *Suppose we have $\|X' - X\|_\infty \leq \epsilon$.*

- *Let $R > 1$.*

- *Assume the value of each entry in matrices can be bounded by $R$.*

- *Let $\mathsf{AAttC}$ denote the approximated attention layer defined in Definition 4.2.*

- Let $U, V \in \mathbb{R}^{n \times k}$ represent low-rank matrices constructed for polynomial approximation of attention matrix $\mathsf{AAttC}(X)$.

- Let $f$ be a polynomial with degree $g$.

*Then, we can show that*

$$\|\mathsf{AAttC}(X') - \mathsf{AAttC}(X)\|_\infty \leq O(kR^{g+2}d) \cdot \epsilon$$

*Proof.* Firstly, given $X$ and $X'$, we need to compute $Q, Q', K, K'$. And we demonstrate that

$$\begin{aligned}
\|Q - Q'\|_\infty &= \|X \cdot W_Q - X' \cdot W_Q\|_\infty \\
&= \|\underbrace{(X - X')}_{n \times d} \cdot \underbrace{W_Q}_{d \times d}\|_\infty \\
&\leq d \cdot \|X - X'\|_\infty \cdot \|W_Q\|_\infty \\
&\leq R \cdot d \cdot \epsilon \qquad (3)
\end{aligned}$$

The initial step is derived from the computation of matrix $Q$. The second step is a consequence of basic algebra, the third step arises from standard matrix multiplication, and the final step is a result of the condition $|X - X'|_\infty \leq \epsilon$ and the fact that each entry is bounded by $R$.

In the same way, we can have $\|K - K'\|_\infty \leq d \cdot \epsilon \cdot R$.

Then, we move forward to calculate $U \in \mathbb{R}^{n \times k}$ and $V \in \mathbb{R}^{n \times k}$. Specifically, for every $i \in [n]$ and $j \in [k]$, we have $U_{i,j} = f(Q_{i,1}, \ldots, Q_{i,d})$ and $V_{i,j} = f(K_{i,1}, \ldots, K_{i,d})$. Then, we can show that

$$\begin{aligned}
\|U - U'\|_\infty &\leq O(R^{g-1}) \cdot \|Q - Q'\|_\infty \\
&\leq O(R^g d) \cdot \epsilon
\end{aligned}$$

The first step above is derived from Lemma C.1 and the condition that each entry is bounded by $R$, while the second step results from Eq. (3) and basic algebra.

In the same way, we can have $\|V - V'\|_\infty \leq O(R^g d) \cdot \epsilon$.

Finally, we can move forward to calculate $\mathsf{AAttC}(X') = U'V'^\top$ and $\mathsf{AAttC}(X) = UV^\top$. Then, for each $i \in [n], j \in [n]$, it can be demonstrated that

$$\begin{aligned}
|\mathsf{AAttC}(X')_{i,j} - \mathsf{AAttC}(X)_{i,j}| &= |\langle U'_{i,*}, V'_{*,j} \rangle - \langle U_{i,*}, V_{*,j} \rangle| \\
&\leq 2k \cdot O(R^g d) \cdot \epsilon \cdot R \\
&\leq O(kR^{g+1}d) \cdot \epsilon
\end{aligned}$$

The first step above is a result of basic algebra, the second step comes from Lemma C.2 and the lemma's condition, and the final step is derived from basic algebra.

Thus, using the definition of the $\ell_\infty$ norm of a matrix, we can demonstrate that

$$\|\mathsf{AAttC}(X') - \mathsf{AAttC}(X)\|_\infty \leq O(kR^{g+1}d) \cdot \epsilon$$

Thus, we complete the proof. $\qquad \square$

## C.4 ERROR ANALYSIS OF $\mathsf{AAttC}(X')$ AND $\mathsf{Attn}(X)$

In this section, we conduct the error analysis between $\mathsf{AAttC}(X')$ and $\mathsf{Attn}(X)$.

**Lemma C.4** (Error analysis of $\mathsf{AAttC}(X')$ and $\mathsf{Attn}(X)$)**.** *Assuming the conditions below are satisfied:*

- *Let $X \in \mathbb{R}^{n \times d}$ denote the input matrix.*

- *Define $X' \in \mathbb{R}^{n \times d}$ as the approximation version of input matrix.*

- *Let $\epsilon \in (0, 0.1)$ denote the approximation error.*

- *Suppose we have $\|X' - X\|_\infty \leq \epsilon$.*

- *Let $R > 1$.*

- *Assume the value of each entry in matrices can be bounded by $R$.*

- *Let $\mathsf{Attn}$ denote the attention layer defined in Definition 3.4.*

- *Let $\mathsf{AAttC}$ denote the approximated attention layer defined in Definition 4.2.*

- *Let $U, V \in \mathbb{R}^{n \times k}$ be low-rank matrices constructed for polynomial approximation of attention matrix $\mathsf{AAttC}(X)$.*

- *Let $f$ be a polynomial with degree $g$.*

*We can demonstrate the following:*

$$\|\mathsf{AAttC}(X') - \mathsf{Attn}(X)\|_\infty \leq O(kR^{g+1}d) \cdot \epsilon$$

*Proof.* It can be shown that

$$
\begin{aligned}
\|\mathsf{AAttC}(X') - \mathsf{Attn}(X)\|_\infty &= \|(\mathsf{AAttC}(X') - \mathsf{AAttC}(X)) + (\mathsf{AAttC}(X) - \mathsf{Attn}(X))\|_\infty \\
&\leq \|(\mathsf{AAttC}(X') - \mathsf{AAttC}(X))\|_\infty + \|(\mathsf{AAttC}(X) - \mathsf{Attn}(X))\|_\infty \\
&\leq O(kR^{g+1}d) \cdot \epsilon + \epsilon \\
&= O(kR^{g+1}d) \cdot \epsilon
\end{aligned}
$$

The first step is based on simple algebra, the second step is derived using the triangle inequality, the third step is obtained from Lemma C.3 and Lemma E.2, and the final step results from basic algebra.

$\square$

### C.5 ERROR ANALYSIS OF UP-INTERPOLATION LAYER

Furthermore, we still need the error analysis of up-interpolation layers.

**Lemma C.5** (Error Analysis of Up-Interpolation Layer). *If the following conditions hold:*

- *Let $X \in \mathbb{R}^{h \times w \times d}$ denote the input feature map.*

- *Define $X' \in \mathbb{R}^{h \times w \times d}$ as the approximated input feature map.*

- *Let $\Phi_{\mathrm{up}} : \mathbb{R}^{h \times w \times d} \to \mathbb{R}^{h' \times w' \times d}$ represent the pyramid up-interpolation layer defined in Definition 3.3.*

- *Let $W : \mathbb{R} \to \mathbb{R}$ be a bicubic spline kernel as defined in 3.1.*

- *Let $\epsilon \in (0, 0.1)$ denote the approximation error.*

- *Let $\|X - X'\|_\infty \leq \epsilon$.*

*Then we have*

$$\|\Phi_{\mathrm{up}}(X) - \Phi_{\mathrm{up}}(X')\|_\infty \leq O(\epsilon)$$

*Proof.* For each $i \in [h'], j \in [w'], l \in [d]$, we have

$$
\begin{aligned}
|\Phi_{\mathrm{up}}(X)_{i,j,l} - \Phi_{\mathrm{up}}(X')_{i,j,l}| &= |\sum_{s=-1}^{2} \sum_{t=-1}^{2} W(s) \cdot (X_{\frac{ih}{h'}+s, \frac{jw}{w'}+t, l} - X'_{\frac{ih}{h'}+s, \frac{jw}{w'}+t, l}) \cdot W(t)| \\
&\leq \sum_{s=-1}^{2} \sum_{t=-1}^{2} |W(s) \cdot (X_{\frac{ih}{h'}+s, \frac{jw}{w'}+t, l} - X'_{\frac{ih}{h'}+s, \frac{jw}{w'}+t, l}) \cdot W(t)|
\end{aligned}
$$

$$\leq \sum_{s=-1}^{2} \sum_{t=-1}^{2} |W(s) \cdot W(t)| \cdot \epsilon$$
$$= O(\epsilon)$$

The first step is based on Definition 3.2, the second step is derived using the triangle inequality, the third step is a consequence of $\|X - X'\|_\infty \leq \epsilon$, and the final step follows from $W(x) \in [0, 1]$ and basic algebra.

Then, according to the definition of the $l_\infty$ norm, we obtain

$$\|\Phi_{\mathrm{up}}(X) - \Phi_{\mathrm{up}}(X')\|_\infty \leq O(\epsilon)$$

$\square$

### C.6 ERROR ANALYSIS FOR VAR TRANSFORMER

Then, we move forward to show the error propagation analysis for one VAR Transformer Layer.

**Lemma C.6** (Error propagation analysis for one VAR Transformer Layer). *Assuming the conditions below are satisfied:*

- *Let $X \in \mathbb{R}^{n \times d}$ denote the input data matrix.*

- *Define $X' \in \mathbb{R}^{n \times d}$ as the approximation version of $X$.*

- *Let $\epsilon \in (0, 0.1)$ denote the approximation error.*

- *Suppose we have $\|X' - X\|_\infty \leq \epsilon$.*

- *Let $R > 1$.*

- *Assume the value of each entry in matrices can be bounded by $R$.*

- *Let* Attn *denote the attention layer defined in Definition 3.4.*

- *Let* AAttC *denote the approximated attention layer defined in Definition 4.2.*

- *Let $U, V \in \mathbb{R}^{n \times k}$ be low-rank matrices constructed for polynomial approximation of attention matrix* AAttC$(X)$.

- *Let $f$ be a polynomial with degree $g$.*

*It can be shown that*

$$\|\mathsf{AAttC}(\Phi_{\mathrm{up}}(X')) - \mathsf{Attn}(\Phi_{\mathrm{up}}(X))\|_\infty \leq O(kR^{g+1}d) \cdot \epsilon$$

*Proof.* The result is easily derived from Lemma C.4 and Lemma C.5. $\square$

### C.7 ERROR ANALYSIS FOR THE FAST VAR TRANSFORMER

We perform an error analysis of the Fast VAR Transformer in this section.

**Lemma C.7** (Error analysis of the Fast VAR Transformer, formal version of Lemma 5.4). *If the following conditions hold:*

- *Let $X_{\mathrm{init}} \in \mathbb{R}^{1 \times d}$ denote the initial input token map.*

- *Let $K \in \mathbb{N}$ represent the number of approximate attention layers in the* VAR *model.*

- *Let the* VAR *transformer* $\mathsf{TF}_K$ *be defined as Definition 3.5.*

- *Let the Fast* VAR *Transformer* $\mathsf{FTF}_K$ *as given in Definition 4.3.*

- *For $i \in [K]$, let $\mathsf{TF}_i(X_{\mathrm{init}})$ denote the output of the $i$-th iteration of the* VAR *transformer.*

- *For $i \in [K]$, let $\mathsf{FTF}_i(X_{\mathrm{init}})$ denote the output of the $i$-th iteration of the fast VAR transformer.*

- *Let $\mathsf{TF}_K(X_{\mathrm{init}}) \in \mathbb{R}^{O(n^2) \times d}$ denote the final output of the VAR transformer.*

- *Let $\mathsf{FTF}_K(X_{\mathrm{init}}) \in \mathbb{R}^{O(n^2) \times d}$ denote the final output of the fast VAR transformer.*

- *Assume each entry in the matrices can be represented using $O(\log n)$ bits.*

- *Let $U, V \in \mathbb{R}^{n \times k}$ be low-rank matrices constructed for polynomial approximation of attention matrix $\mathsf{AAttC}(X)$.*

- *Let $f$ be a polynomial with degree $g$.*

*Then, we can show that the error bound of the final output $\mathsf{FTF}_K(X_{\mathrm{init}})$ as*

$$\|\mathsf{FTF}_K(X_{\mathrm{init}}) - \mathsf{TF}_K(X_{\mathrm{init}})\|_\infty \leq 1/\operatorname{poly}(n)$$

*Proof.* We can conduct math induction as the following:

Consider the first iteration. We can show that.

$$\|\mathsf{FTF}_1(X_{\mathrm{init}}) - \mathsf{TF}_1(X_{\mathrm{init}})\|_\infty = \|\mathsf{AAttC}_1(X_{\mathrm{init}}) - \mathsf{Attn}_1(X_{\mathrm{init}})\|_\infty$$
$$\leq 1/\operatorname{poly}(n)$$

The first step is based on Definition 3.5 and Definition 4.3, and the final step follows from Lemma E.2.

Assume that the following statement is true for the $k$-th iteration (where $k < K$):

$$\|\mathsf{FTF}_k(X_{\mathrm{init}}) - \mathsf{TF}_k(X_{\mathrm{init}})\|_\infty \leq 1/\operatorname{poly}(n) \tag{4}$$

Then we move forward to consider the $k+1$-th iteration as the following:

$$\|\mathsf{FTF}_{k+1}(X_{\mathrm{init}}) - \mathsf{TF}_{k+1}(X_{\mathrm{init}})\|_\infty$$
$$= \|\mathsf{AAttC}_{k+1}(\Phi_{\mathrm{up},k}(\mathsf{FTF}_k(X_{\mathrm{init}}))) - \mathsf{Attn}_{k+1}(\Phi_{\mathrm{up},k}(\mathsf{TF}_k(X_{\mathrm{init}})))\|_\infty$$
$$\leq 1/\operatorname{poly}(n)$$

The first step is based on Definition 3.5 and Definition 4.3, the second step is derived from Lemma C.6, the fact that each entry in the matrices can be represented using $O(\log(n))$ bits, and Eq. (4).

Finally, we can use math induction to show that

$$\|\mathsf{FTF}_K(X_{\mathrm{init}}) - \mathsf{TF}_K(X_{\mathrm{init}})\|_\infty \leq 1/\operatorname{poly}(n)$$

Thus, we complete the proof. $\square$

## D  ERROR ANALYSIS OF VQVAE DECODER

In this section, we conduct the error analysis of the VQ-VAE Decoder. Firstly, the following lemma presents the error analysis of the Convolution Layer.

**Lemma D.1** (Error analysis of Convolution Layer). *Assuming the conditions below are satisfied:*

- *Let $X \in \mathbb{R}^{h \times w \times c_{\mathrm{in}}}$ denote the input feature map.*

- *Let $X' \in \mathbb{R}^{h \times w \times c_{\mathrm{out}}}$ denote the output feature map.*

- *Let $\phi_{\mathrm{conv}} : \mathbb{R}^{h \times w \times c_{\mathrm{in}}} \to \mathbb{R}^{h \times w \times c_{\mathrm{out}}}$ denote the convolution layer defined in Definition 3.6.*

- *Let $\epsilon \in (0, 0.1)$ denote the approximation error.*

- *Let $\|X - X'\|_\infty \leq \epsilon$.*

- *Let $R > 1$.*

- *Assume the value of each entry in matrices can be bounded by $R$.*

- *Let $C = 9c_{\mathrm{in}}$ denote a constant.*

*Then we have*

$$\|\phi_{\mathrm{conv}}(X) - \phi_{\mathrm{conv}}(X')\|_\infty \leq C\epsilon R$$

*Proof.* For each $i \in [h], j \in [w], l \in [c_{\mathrm{out}}]$, we have

$$
\begin{aligned}
|\phi_{\mathrm{conv}}(X)_{i,j,l} - \phi_{\mathrm{conv}}(X')_{i,j,l}| &= |\sum_{m=1}^{3}\sum_{n=1}^{3}\sum_{c=1}^{c_{\mathrm{in}}}(X_{i+m-1,j+n-1,c} - X'_{i+m-1,j+n-1,c}) \cdot K^l_{m,n,c}| \\
&\leq \sum_{m=1}^{3}\sum_{n=1}^{3}\sum_{c=1}^{c_{\mathrm{in}}}|(X_{i+m-1,j+n-1,c} - X'_{i+m-1,j+n-1,c}) \cdot K^l_{m,n,c}| \\
&\leq \sum_{m=1}^{3}\sum_{n=1}^{3}\sum_{c=1}^{c_{\mathrm{in}}}\epsilon \cdot R \\
&\leq 9 \cdot c_{\mathrm{in}}\epsilon R \\
&= C\epsilon R
\end{aligned}
$$

The 1st step is a consequence of Definition 3.6, the 2nd step is based on the triangle inequality, the 3rd is a result of the lemma's conditions, the fourth arises from elementary algebra, and the final step stems from the definition of $C$. $\qquad\square$

Then, we can show the lemma, which presents the error analysis of the Fast VQ-VAE Decoder.

**Lemma D.2** (Error analysis of Fast VQ-VAE Decoder ). *In the case that the following conditions are satisfied:*

- *Let $X \in \mathbb{R}^{n \times d}$ denote the input matrix.*

- *Let the up-interpolation Layer $\phi_{\mathrm{up}}$ be defined as Definition 3.2.*

- *Let the convolution layer $\phi_{\mathrm{conv}}$ be defined as Definition 3.6.*

- *Let the attention layer $\mathsf{Attn}$ be defined as Definition 3.4*

- *Let the fast attention layer $\mathsf{AAttC}$ be defined as Definition 4.2.*

- *Let the VQ-VAE Decoder be the composition of a constant number of up-interpolation layers, convolutions layers, and attention layers.*

- *Let the Fast VQ-VAE Decoder be defined as substituting all $\mathsf{Attn}$ layers in VQ-VAE with $\mathsf{AAttC}$ layers.*

*Then, we can show that the approximation error of the Fast VQ-VAE Decoder can be bounded by $1/\operatorname{poly}(n)$.*

*Proof.* By Lemma C.6, we have shown that fast attention computation will introduce an approximation error no more than $1/\operatorname{poly}(n)$.

Then, by Lemma C.5 and Lemma D.1, we still can bounded have shown the approximation error by $1/\operatorname{poly}(n)$ after passing another up-interpolation layer, convolutions layer.

Since VQ-VAE is a composition of a constant number of up-interpolation layers, convolution layers, and attention layers, the overall approximation error can still be bounded by $1/\operatorname{poly}(n)$. $\qquad\square$

# E   RUNNING TIME

In this section, we conduct the running time analysis of every component of the VAR model and the fast VAR model. In Section E.1, we conduct the running time analysis of the VAR transformer and the fast transformer. In Section E.2, we conduct the running time analysis of the feature map reconstruction layer. In Section E.3, we conduct the running time analysis of the VQVAE Decoder and fast VQVAE Decoder.

## E.1   PHASE 1: RUNNING TIME OF TOKEN MAPS GENERATION

In this section, we present lemmas on the time complexity of VAR transformer defined in Definition 3.5 and fast VAR transformer defined in Definition 4.3.

**Lemma E.1** (Running time of VAR Transformer). *If the following conditions hold:*

- *Assume the* VAR *transformer defined in Definition 3.5 has $K$ attention layers.*

- *Let $k_1 \in [K]$ and $k_2 \in [K-1]$.*

- *Let $X_{\mathrm{init}} \in \mathbb{R}^{1 \times 1 \times d}$ denote the first scale token map.*

- *Let $\alpha > 1$ denote the growth rate of the height and width of the token map at each level. Then for $k_1 \in [K]$, the $k_1$-th token map $r_{k_1} \in \mathbb{R}^{\alpha^{k_1-1} \times \alpha^{k_1-1} \times d}$.*

- *Let $r_K \in \mathbb{R}^{n \times n \times d}$ denote the last scale token map, where $n = \alpha^{K-1}$.*

- *Let $\Phi_{\mathrm{up},k_2}$ denote the $k_2$-th pyramid up-interpolation layer defined in Def 3.3.*

- *Let $\mathsf{Attn}_{k_1}$ denote the $k_1$-th attention layer defined in Definition 3.4.*

- *Let $d = O(\log(n))$ denote the embedding size of each token.*

*then the time complexity of* VAR *transformer* TF *is $O(n^{4+o(1)})$.*

*Proof.* The runtime computation of the VAR transformer can be divided into two components: the runtime of the Attention layers $\mathsf{Attn}_1, \ldots, \mathsf{Attn}_K$ and the runtime of the pyramid up-interpolation layer $\Phi_{\mathrm{up},1}, \ldots, \Phi_{\mathrm{up},K-1}$.

**Part 1.** Running time of the attention layers $\mathsf{Attn}_1, \ldots, \mathsf{Attn}_K$.

Firstly, we consider the $k_1$-th autoregressive token map generation. We use $L_{k_1}$ to denote the total number of the tokens input into the VAR attention layer at $k_1$-th step and $L_{k_1}$ can be calculated as the following:

$$
\begin{aligned}
L_{k_1} &= \sum_{i=1}^{k_1} (\alpha^{i-1})^2 \\
&= \frac{\alpha^{2k_1} - 1}{\alpha^2 - 1} \\
&\leq \frac{\alpha^{2k_1}}{\alpha^2 - 1} \\
&\leq \frac{\alpha^{2k_1}}{0.5\alpha^2} \\
&= 2 \cdot \alpha^{2k_1 - 2}
\end{aligned}
\tag{5}
$$

In the first step, we use the condition of this lemma. The second and third steps are a consequence of basic algebra. The fourth step is due to $\alpha \geq 2$, and the last step is derived from elementary algebra.

Thus, at the $k_1$-th step, we use $X_{k_1} \in \mathbb{R}^{L_{k_1} \times d}$ to denote the input matrix. The attention computation cost at the $k_1$-th step is $O(L_{k_1}^2 d)$. We then sum up the computation time across all $K$ steps:

$$
\mathcal{T}_{\mathrm{attn}} = O((L_1^2 + L_2^2 + \cdots + L_K^2) \cdot d)
$$

$$\leq O\Big( \sum_{k_1=1}^{(\log_\alpha n)+1} (2 \cdot \alpha^{2k_1-2})^2 \cdot d \Big)$$

$$= O\Big( \sum_{k_1=1}^{(\log_\alpha n)+1} 4\alpha^{4k_1-4} \cdot d \Big)$$

$$= O(n^4 d)$$

$$= O(n^{4+o(1)})$$

In the first step, the total time is calculated by summing the attention computation times for each $k_1$-th step. The second step follows directly from Eq. (5), while the third and fourth step is a result of basic algebraic manipulation. The last step follows from the condition that $d = O(\log(n))$.

**Part 2.** Running time of the pyramid up-interpolation layers $\Phi_{\mathrm{up},1}, \ldots, \Phi_{\mathrm{up},K-1}$.

We begin by considering the runtime of the last pyramid up-interpolation layer, $\Phi_{\mathrm{up},K-1}$. From Definition 3.5, we know that the output of $\Phi_{\mathrm{up},K-1}$ serves as the input to $\mathsf{Attn}_K$. Therefore, the number of tokens generated by $\Phi_{\mathrm{up},K-1}$ is $L_K \leq 2n^2$, with the inequality following from Eq. 5. Furthermore, by Eq. (1), we know that every token generated by $\Phi_{\mathrm{up},K-1}$ needs $O(d)$ times multiplications and $O(d)$ times additions. Thus, the running time of $\Phi_{\mathrm{up},K-1}$ is

$$\mathcal{T}_{\mathrm{up}}^{K-1} \leq O(d) \cdot L_K$$
$$= O(n^2 d) \tag{6}$$

where the step comes from summing up the time cost for each token generated by $\Phi_{\mathrm{up},K-1}$.

For each $k' \in [K-2]$, the number of tokens generated by $\Phi_{\mathrm{up},k'}$ is less than $L_K$ which is due to Definition 3.3. Then we can compute the total rumming time for pyramid up-interpolation layers $\Phi_{\mathrm{up},1}, \ldots, \Phi_{\mathrm{up},K-1}$:

$$\mathcal{T}_{\mathrm{up}} \leq (K-1) \cdot \mathcal{T}_{up}^{K-1}$$
$$= O(\log(n)) \cdot O(n^2 d)$$
$$= O(n^{2+o(1)})$$

where the step comes from summing up the running time of $\Phi_{\mathrm{up},1}, \ldots, \Phi_{\mathrm{up},K-1}$, the second step follows from $n = \alpha^{K-1}$ and Eq (6) and the last step follows from $d = O(\log(n))$ and simple algebra.

Thus, by summing up the $\mathcal{T}_{\mathrm{attn}}$ and $\mathcal{T}_{\mathrm{up}}$, we know that the time complexity of VAR transformer is $O(n^{4+o(1)})$.

$$\square$$

Then, we show a lemma that demonstrates a fast way to compute attention in (Alman & Song, 2023).

**Lemma E.2** (Fast Attention Computation, Theorem 1.4 of (Alman & Song, 2023)). *Let* AAttC *be defined as Definition 4.2. Then we have* $\mathsf{AAttC}(n, d = O(\log n), R = \Theta(\sqrt{\log n}), \delta = 1/\operatorname{poly}(n))$ *can be solved in time* $\mathcal{T}_{\mathrm{mat}}(n, n^{o(1)}, d) = n^{1+o(1)}$.

Now we can apply the result Lemma E.2 to the VAR Transformer.

**Lemma E.3** (Running time of Fast VAR Transformer, formal version of Lemma 5.1). *Assuming the conditions below are satisfied:*

- *Assume the fast* VAR *transformer defined in Definition 4.3 has $K$ attention layers.*

- *Let $k_1 \in [K]$ and $k_2 \in [K-1]$.*

- *Let $X_{\mathrm{init}} \in \mathbb{R}^{1 \times 1 \times d}$ denote the first scale token map.*

- *Let $\alpha > 1$ denote the growth rate of the height and width of the token map at each level. Then for $k_1 \in [K]$, the $k_1$-th token map $r_{k_1} \in \mathbb{R}^{\alpha^{k_1-1} \times \alpha^{k_1-1} \times d}$.*

- *Let $r_K \in \mathbb{R}^{n \times n \times d}$ denote the last scale token map, where $n = \alpha^{K-1}$.*

- *Let $d = O(\log n)$ denote the embedding size of each token.*

- *Let $\Phi_{\mathrm{up},k_2}$ denote the $k_2$-th pyramid up-interpolation layer defined in Def 3.3.*

- *Let $\mathsf{AAttC}_{k_1}$ denote the $k_1$-th approximate attention layer defined in Definition 4.2.*

*Then, the total running time of the fast VAR transformer FTF can be accelerated to $O(n^{2+o(1)})$.*

*Proof.* The runtime computation of the fast VAR transformer can be divided into two components: the runtime of the approximate attention layers $\mathsf{AAttC}_1, \dots, \mathsf{AAttC}_K$ and the runtime of the pyramid up-interpolation layer $\Phi_{\mathrm{up},1}, \dots, \Phi_{\mathrm{up},K-1}$.

**Part 1.** Running time of the attention layers $\mathsf{AAttC}_1, \dots, \mathsf{AAttC}_K$.

To generate the $k$-th token map, let the input be $X \in \mathbb{R}^{L_k \times d}$. And we have

$$L_k \leq 2 \cdot \alpha^{2k-2} \tag{7}$$

where this step is a consequence of Eq. (5). So the transformer computation cost at the $k$-th step can be improved from $O(L_k^2 d)$ to $O(L_k^{1+o(1)} d)$ by using the result of Lemma E.2. Then, we sum up the computation time across all $K$ steps:

$$
\begin{aligned}
\mathcal{T}_{\mathrm{aattc}} &= O((L_1 + L_2 + \cdots + L_K) \cdot d) \\
&\leq O\left( \sum_{k=1}^{\log_\alpha n + 1} (2 \cdot \alpha^{2k-2})^{1+o(1)} \cdot d \right) \\
&= O(n^{2+o(1)} d) \\
&= O(n^{2+o(1)})
\end{aligned}
$$

where the first step follows from summing up the computation time of $\mathsf{AAttC}_1, \dots, \mathsf{AAttC}_K$, the second step follows from Eq. (7), the third step follows from simple algebra and the last step follows from the condition that $d = \log(n)$.

**Part 2.** Running time of the pyramid up-interpolation layers $\Phi_{\mathrm{up},1}, \dots, \Phi_{\mathrm{up},K-1}$.

The total running time for pyramid up-interpolation layers $\Phi_{\mathrm{up},1}, \dots, \Phi_{\mathrm{up},K-1}$ is the same as Lemma E.1:

$$\mathcal{T}_{\mathrm{up}} = O(n^{2+o(1)})$$

Thus, by summing up the $\mathcal{T}_{\mathrm{aattc}}$ and $\mathcal{T}_{\mathrm{up}}$, we know that the time complexity of VAR transformer is $O(n^{2+o(1)})$.

$\square$

## E.2 PHASE 2: RUNNING TIME OF FEATURE MAP RECONSTRUCTION

In this section, we analyze the total runtime of the VAR models for feature map reconstruction.

**Lemma E.4** (Running time of Feature Map Reconstruction Layer, formal version of Lemma 5.2). *If the following conditions hold:*

- *Let $K \in \mathbb{N}$ denote the total number of the token maps.*

- *Let $k \in [K]$.*

- *Let $d$ denote the embedding size of each token.*

- *Let $X_{\mathrm{init}} \in \mathbb{R}^{1 \times 1 \times d}$ denote the initial token map.*

- *Let $\alpha > 1$ denote the growth rate of the height and width of the token map at each level. Then for $k \in [K]$, the $k$-th token map $r_k \in \mathbb{R}^{\alpha^{k-1} \times \alpha^{k-1} \times d}$.*

- *Let $r_K \in \mathbb{R}^{n \times n \times d}$ denote the last scale token map, where $n = \alpha^{K-1}$.*

*then the total runtime of the* VAR *models for feature map reconstruction is $O(n^{2+o(1)})$.*

*Proof.* For each $k \in [K]$, VAR Model will up-interpolate token map $r_k \in \mathbb{R}^{\alpha^{k-1} \times \alpha^{k-1} \times d}$ to $r'_k \in \mathbb{R}^{n \times n \times d}$ by using bicubic interpolation defined in Definition 3.2. Specifically, the computation of each token in $r'_k$ requires $O(d)$ multiplications and $O(d)$ additions which is due to Eq. (1). Thus, the computation cost for the up-interpolation of each token map is

$$\mathcal{T}_{\text{up}}^k = O(d) \cdot n^2$$
$$= O(n^2 d)$$

There are total $K = O(\log n)$ token maps needed to be up-interpolated, so the total time for up-interpolation is

$$\mathcal{T}_{\text{up}} = O(n^2 d) \cdot O(\log n)$$
$$= O(n^2 d \log n)$$

where the first step follows from summing up the running time of $K$ scale token maps, and the second step follows from simple algebra.

Furthermore, to address the information loss in the up-interpolation process, the VAR Model uses a convolution operation $\phi_{\text{conv}}(\cdot)$ on the token map $\{r'_1, \ldots, r'_K\}$ generated by up-interpolation. We assume the convolution kernel size is $3 \times 3 \times d$, and the convolution layer does not change the dimension of each token map, i.e., for each $i \in [K]$, $\phi(r'_i) \in \mathbb{R}^{n \times n \times d}$. Hence, for every entry in $\phi(r'_i)$, it needs $O(d)$ operations. Then, we can have the convolution computation time for one token map is

$$\mathcal{T}_{\text{conv}}^k = O(d) \cdot n^2 d$$
$$= O(n^2 d^2)$$

In the first step, the total computation time is obtained by adding the times for the $n \times n \times d$ entries, while the second step results from simple algebra.

There are total $O(\log n)$ token maps needed to be passed to the convolution layer, so the total time for convolution operations is

$$\mathcal{T}_{\text{conv}} = O(\log n) \cdot O(n^2 d^2)$$
$$= O(n^2 d^2 \log n)$$

Then, the VAR Model will sum up $O(\log n)$ token maps processed by convolution layers, where each token map has a size of $n \times n \times d$. Thus, the computation cost of addition needs

$$\mathcal{T}_{\text{add}} = O(\log n) \cdot (n^2 d)$$
$$= O(n^2 d \log n)$$

In the first step, token maps are added element-wise, and there are $O(\log(n))$ token maps in total, while the second step results from basic algebra.

Hence, the running time of feature map reconstruction is as follows:

$$\mathcal{T}_{\text{rc}} = \mathcal{T}_{\text{up}} + \mathcal{T}_{\text{conv}} + \mathcal{T}_{\text{add}}$$
$$= O(n^2 d^2 \log n)$$
$$= O(n^{2+o(1)})$$

The first step is derived by summing the times for up-interpolation operations, convolution operations, and token map additions, while the second step is due to basic algebra. The last step follows from $d = O(\log n)$. $\square$

### E.3 PHASE 3: RUNNING TIME OF VQ-VAE DECODER

In this section, we analyze the running time of the VQ-VAE Decoder and fast VQ-VAE Decoder.

**Lemma E.5** (Running time of VQ-VAE Decoder). *If the following conditions hold:*

- *Let $k_1, k_2, k_3 \in \mathbb{N}$ be constant numbers.*

- *Given $X \in \mathbb{R}^{n \times n \times d}$ as the input feature map.*

- *Let $d = O(\log n)$*

- *Assume that there are $k_1$ up-interpolation layers $\phi_{\mathrm{up}}$ defined in Definition 3.2.*

- *Given a feature map $M \in \mathbb{R}^{h \times w \times d}$. For $i \in [k_1]$, we assume $i$-th up-interpolation layer's output $\phi_{\mathrm{up}}^i(M) \in \mathbb{R}^{O(h) \times O(w) \times d}$.*

- *We assume there are $k_2$ attention layers $\mathsf{Attn}$ defined in Definition 3.4.*

- *Given a feature map $M \in \mathbb{R}^{h \times w \times d}$. For $i \in [k_1]$, the $i$-th attention layer's output $\mathsf{Attn}(M) \in \mathbb{R}^{h \times w \times d}$.*

- *We assume there are $k_3$ convolution layers $\phi_{\mathrm{conv}}$ defined in Definition 3.6.*

- *Given a feature map $M \in \mathbb{R}^{h \times w \times d}$. For $i \in [k_1]$, we assume $i$-th convolution layer's output $\phi_{\mathrm{conv}}^i(M) \in \mathbb{R}^{h \times w \times O(d)}$.*

*then the total running time of the VQ-VAE Decoder is $O(n^{4+o(1)})$.*

*Proof.* By the condition, we can have that for each $l \in [k_1 + k_2 + k_3]$, the size of the output $M^l$ of any intermediate layer (up-interpolation layer, convolution layer, attention layer) is $O(n) \times O(n) \times O(d)$. The running time can be computed as follows:

**Part 1. Running time of Up-interpolation layers.** For each $l \in [k_1]$, we assume $M^l \in \mathbb{R}^{O(n) \times O(n) \times O(d)}$ as the output feature map from the $l$-th up-interpolation layer. For every token of $M^l$, it needs $O(d)$ multiplications and $O(d)$ additions (see more details in Definition 3.2). Thus, the computation cost for the feature map $M^l$ is

$$
\begin{aligned}
\mathcal{T}_{\mathrm{up}}^l &= O(d) \cdot O(n) \cdot O(n) \\
&= O(n^2 d) \\
&= O(n^{2+o(1)})
\end{aligned}
$$

The first step is derived by summing the computation costs for each entry of $M^l$, while the second step is due to basic algebra. The last step follows from the condition that $d = O(\log n)$.

Since there are $k_1$ up-interpolation layers in total, the total time of the up-interpolation layers in the VQ-VAE decoder is

$$
\begin{aligned}
\mathcal{T}_{\mathrm{up}} &= k_1 \cdot O(n^{2+o(1)}) \\
&= O(n^{2+o(1)})
\end{aligned}
\tag{8}
$$

The first step is derived by summing the computation costs for each up-interpolation layer, while the second step is due to the fact that $k_1$ is a constant number.

**Part 2. Running time of Attention layers.** For each $l \in [k_2]$, we assume $M^{l-1} \in \mathbb{R}^{O(n) \times O(n) \times O(d)}$ as the feature map used as input for the $l$-th attention layer. We can consider the input of this attention layer as a sequence with length $O(n^2)$ and embedding dimension $O(d)$. Hence, the computation cost of this attention layer is

$$
\begin{aligned}
\mathcal{T}_{\mathrm{attn}}^l &= O(n^4 d) \\
&= O(n^{4+o(1)})
\end{aligned}
$$

where the first step follows from computation cost for standard attention computation, the second step follows from the condition $d = O(\log n)$.

Since there are $k_2$ attention layers in total, the total time of the attention layers in the VQ-VAE decoder is

$$\mathcal{T}_{\text{attn}} = k_2 \cdot O(n^{4+o(1)})$$
$$= O(n^{4+o(1)})$$

The first step is derived by summing the computation costs for each attention layer, while the second step is due to the fact that $k_2$ is a constant.

**Part 3.    Running time of Convolution layers.**    For each $l \in [k_3]$, we assume $M^l \in \mathbb{R}^{O(n) \times O(n) \times O(d)}$ as the output feature map of the $l$-th convolution layer. For every entry of $M^l$, it needs $O(d)$ operations. Thus, the computation cost for the feature map $M^l$ is

$$\mathcal{T}_{\text{conv}}^l = O(d) \cdot O(n^2 d)$$
$$= O(n^2 d^2)$$
$$= O(n^{2+o(1)})$$

The first step is derived by summing the computation costs for each entry of $M^l$, while the second step stems from basic algebra. The last step follows from the condition that $d = O(\log n)$.

Since there are $k_3$ convolution layers in total, the total time of the convolution layers in the VQ-VAE decoder is

$$\mathcal{T}_{\text{conv}} = k_3 \cdot O(n^{2+o(1)})$$
$$= O(n^{2+o(1)}) \tag{9}$$

The first step is derived by summing the computation costs for each convolution layer, while the second step is due to the fact that $k_3$ is a constant.

Finally, the computation cost of the VQ-VAE decoder can be calculated as follows:

$$\mathcal{T}_{\text{dec}} = \mathcal{T}_{\text{up}} + \mathcal{T}_{\text{attn}} + \mathcal{T}_{\text{conv}}$$
$$= O(n^{4+o(1)})$$

The first step results from summing the computation costs of up-interpolation, attention, and convolution layers, while the second step follows from simple algebra.

$\square$

Then, we move forward to show the running time of the fast VQ-VAE decoder.

**Lemma E.6** (Running time of Fast VQ-VAE Decoder, formal version of Lemma 5.3). *If the following conditions hold:*

- *Let $k_1, k_2, k_3 \in \mathbb{N}$ be constant numbers.*

- *Given $X \in \mathbb{R}^{n \times n \times d}$ as the input feature map.*

- *Let $d = O(\log n)$*

- *Assume that there are $k_1$ up-interpolation layers $\phi_{\text{up}}$ defined in Definition 3.2.*

- *Given a feature map $M \in \mathbb{R}^{h \times w \times d}$. For $i \in [k_1]$, we assume $i$-th up-interpolation layer's output $\phi_{\text{up}}^i(M) \in \mathbb{R}^{O(h) \times O(w) \times d}$.*

- *We assume there are $k_2$ approximate attention layers $\mathsf{AAttC}$ defined in Definition 4.2.*

- *Given a feature map $M \in \mathbb{R}^{h \times w \times d}$. For $i \in [k_1]$, the $i$-th approximate attention layer's output $\mathsf{AAttC}(M) \in \mathbb{R}^{h \times w \times d}$.*

- *We assume there are $k_3$ convolution layers $\phi_{\text{conv}}$ defined in Definition 3.6.*

- *Given a feature map $M \in \mathbb{R}^{h \times w \times d}$. For $i \in [k_1]$, we assume $i$-th convolution layer's output $\phi_{\mathrm{conv}}^i(M) \in \mathbb{R}^{h \times w \times O(d)}$.*

*then the total runtime of the VQ-VAE decoder can be accelerated to $O(n^{2+o(1)})$.*

*Proof.* As the same in Eq. (8) and Eq. (9), the computation cost for up-interpolation layers and convolution layers in VQ-VAE decoder still needs $O(n^{2+o(1)}d)$.

For each $l \in [k_2]$, we assume $M^{l-1} \in \mathbb{R}^{O(n) \times O(n) \times O(d)}$ as the input feature map for the $l$-th approximate attention layer. We can consider the input of the attention layer as a sequence with length $O(n^2)$ and embedding dimension $O(d)$. By using the result of Lemma E.2, the computation cost of $M^l$ can be speed up to

$$\mathcal{T}_{\mathrm{attn}}^l = O(n^{2+o(1)}d)$$
$$= O(n^{2+o(1)})$$

where the second step follows from the condition that $d = O(\log n)$.

Since there are $k_2$ attention layers in total, the total computation cost of the attention layers in the VQ-VAE decoder is

$$\mathcal{T}_{\mathrm{attn}} = k_2 \cdot O(n^{2+o(1)})$$
$$= O(n^{2+o(1)})$$

The computation cost in the first step is obtained by adding the costs of the up-interpolation layers, attention layers, and convolution layers, while the second step stems from $k_2$ is a constant.

Thus, the total runtime of the VQ-VAE decoder can be calculated as follows:

$$\mathcal{T}_{\mathrm{dec}} = \mathcal{T}_{\mathrm{up}} + \mathcal{T}_{\mathrm{attn}} + \mathcal{T}_{\mathrm{conv}}$$
$$= O(n^{2+o(1)})$$

The computation cost in the first step is obtained by adding the costs of the up-interpolation layers, attention layers, and convolution layers, while the second step comes from simple algebra. $\square$

## LLM USAGE DISCLOSURE

LLMs were used only to polish language, such as grammar and wording. These models did not contribute to idea creation or writing, and the authors take full responsibility for this paper's content.

