# OpenReview forum: "On Computational Limits and Provably Efficient Criteria of Visual Autoregressive Models: A Fine-Grained Complexity Analysis"
_ICLR.cc/2026/Conference — ICLR 2026 Conference Withdrawn Submission_

### Official Review · Reviewer_9je2 · 2025-10-19

**Soundness:** 2
**Presentation:** 3
**Contribution:** 2
**Rating:** 2
**Confidence:** 3

**Summary:**

The authors propose a fine-grained computational analysis of Visual Autoregressive (VAR) models, aiming to uncover the theoretical limits of their computational efficiency. Based on the Strong Exponential Time Hypothesis (SETH), they rigorously prove that achieving sub-quartic runtime \(O(n^{4-\Omega(1)})\) for VAR computations is impossible when \(R = \Theta(\sqrt{\log n})\) and \(d = O(\log n)\) (Theorem 4.5). In contrast, when \(R = o(\sqrt{\log n})\), they construct an almost-quadratic-time algorithm \(O(n^{2+o(1)})\) with only \(1/\mathrm{poly}(n)\) error (Theorem 5.7). Their analysis shows where the theoretical bottleneck lies—in the attention computation—and identifies a precise "phase transition" between infeasible and feasible acceleration.

**Strengths:**

1. Theorem 4.5 provides a clear SETH-based lower bound ruling out sub-quartic algorithms, while Theorem 5.7 offers a constructive upper bound achieving near-quadratic runtime under mild constraints on \(R\). This dual result elegantly establishes a provable boundary between hard and tractable regions for VAR computation.

2.  Section 3 systematically formalizes each VAR component—pyramid up-interpolation, attention, and convolution—before deriving complexity results. Lemmas 5.1–5.3 further decompose the runtime and clarify how the approximate attention layer contributes to the reduced \(O(n^{2+o(1)})\) complexity.

**Weaknesses:**

1. The main results depend on \(d = O(\log n)\) and bounded activation magnitude \(R = \Theta(\sqrt{\log n})\). In real-world VAR systems, hidden dimensions are often thousands, and input norms are unbounded, making it unclear whether the phase transition at $R^* = \sqrt{\log n}$ applies to practical architectures.

2. The reliance on SETH and the Approximate-Attention-Computation (AAttC) reduction anchors the proof in worst-case theoretical hardness. While elegant, it provides little quantitative insight for practitioners about when or how current attention implementations might approach the lower bound.

3. Although Lemmas 5.1–5.6 theoretically describe the fast algorithm, no experimental validation or prototype implementation confirms the predicted asymptotic gain or the $1/\mathrm{poly}(n)$ error guarantee.

**Questions:**

1. In practical setups where \(d\) is large and \(R\) unbounded, how would the phase transition boundary $R^* = \Theta(\sqrt{\log n})$ adapt, and can it meaningfully inform real model scaling?

2. Could the near-quadratic algorithm extend to approximate or kernelized attention methods (such as Nyström or random-feature attention) that already achieve sub-quadratic scaling empirically?

---

> ### Author Response · Authors · 2025-12-01
>
> Thank you for your thoughtful feedback. Your comments are very helpful and much appreciated. We will address these in the next version.

---

### Official Review · Reviewer_mnmt · 2025-10-29

**Soundness:** 2
**Presentation:** 1
**Contribution:** 1
**Rating:** 2
**Confidence:** 4

**Summary:**

This paper studies the computational bounds of visual autoregressive (VAR) models. Since VAR models are composed of upsampling blocks and attention layers, performing full attention incurs $O(n^{4})$ computational cost, where $n$ denotes the height and width of the largest feature map. This paper extends the results of Alman & Song (2023) to VAR models by showing that when the bound of the entries in the attention operation is $R = \Theta(\sqrt{\log n})$, it is impossible to approximate the attention with $1/\mathrm{poly}(n)$ additive error in sub-quadratic time. In contrast, it is possible to approximate the attention with the same additive error under sub-quadratic cost when $R = o(\sqrt{\log n})$ via low-rank attention, consistent with the theoretical framework of Alman & Song (2023).


Alman & Song. "Fast attention requires bounded entries." NeurIPS23

**Strengths:**

The paper provides a rigorous formulation of VAR models and clearly presents the image generation process in detail. Most of definitions are stated precisely and is supported with sufficient explanation.

**Weaknesses:**

I recommend rejection of this paper for the following reasons: (1) the theoretical contributions are insufficient, (2) the paper lacks discussion on whether trained VAR models satisfy the required entry-bound conditions, and (3) the paper is poorly written and insufficiently positioned within the literature.

1. The main theorems are direct applications of the results of Alman & Song (2023), who already analyzed the existence of sub-quadratic approximation algorithms under bounded attention entries. Although the paper includes additional complexity analysis for components of VAR beyond attention layers, the dominant computational bottleneck still arises from attention. The proofs largely rely on a straightforward application of existing results without addressing any technical challenges specific to VAR. Consequently, the paper does not provide novel or meaningful theoretical insights. Given that Alman & Song (2023) already covered the core discussion on approximation with $1/\mathrm{poly}(n)$ additive error, simply extending their results to VAR models is not a sufficiently compelling contribution to justify acceptance.
2. The paper does not establish whether VAR models actually satisfy $R = o(\sqrt{\log n})$ in practice. Because the contribution is predominantly theoretical—and limited, as noted above—the lack of practical justification becomes more problematic. If VAR exhibits unique characteristics that make verifying the entry-bound criterion or implementing low-rank attention more challenging, such aspects should be explained and supported by experiments or additional analysis.
3. The writing includes unnecessary details while omitting important ones, reducing clarity and readability. For example, the Related Work section introduces diffusion models and gradient approximation techniques, which are not central to the paper’s contributions and obscure its positioning within the literature. Conversely, Definition 4.2 only provides a high-level notion of approximate attention without describing implementation details or citing specific low-rank attention methods.

**Questions:**

1. Does a trained VAR model satisfy the bounded-entry criterion? How are the entry bounds change along the pyramid resolution?
2. Lemma E.3 does not assume $R=o(\sqrt{\log{n}})$, which is required to substitute $O(L_k^2d)$ with $O(L_k^{1+o(1)} d)$ based on Lemma E.2. If it was not an oversight, then how is the bounded entry property ensured in this context?

Minor comments:

1. The proof of Theorem 4.5 is missing definitions of $L, a, \alpha,$ and $q$.
2. A typo in Lemma E.2: I believe it should be $R=o(\sqrt{\log{n}})$, not $R=\Theta(\sqrt{\log{n}})$.
3. A typo in Line 334 (above Definition 4.3) “Fa ast” → “Fast”

---

> ### Author Response · Authors · 2025-12-01
>
> Thank you for your thoughtful feedback. Your comments are very helpful and much appreciated. We will address these in the next version.

---

### Official Review · Reviewer_1ckL · 2025-10-31

**Soundness:** 2
**Presentation:** 2
**Contribution:** 2
**Rating:** 4
**Confidence:** 3

**Summary:**

The author proposes a fine-grained complexity analysis of visual autoregressive (VAR) models -- autoregressive image generators that uses a coarse-to-fine next-scale prediction scheme..

Let n be the height/width of the final VQ code map, $d=O(\log n)$ the embedding size, and R an upper bound on entries used in attention. The authors formalize the three-stage VAR pipeline (VAR transformer, feature map reconstruction, VQ-VAE decoder) and prove a sharp phase transition in computational complexity driven by R. The lower bound and upper bound identify a critical boundary $R*=o(\sqrt(\log n))$ and establish when VAR inference is provably fast vs. probably hard, providing the first theory-driven runtime criteria for coarse-to-fine autoregressive image generation.

**Strengths:**

S1. The authors proposed a fine-grained complexity analysis of visual autoregressive (VAR) models.

S2. The authors proposed a conditional lower bound for VAR models.

S3. The authors proved the existence of an algorithm that approximates VAR models in almost quadratic time.

**Weaknesses:**

W1. The contributions are limited due to assumptions lacking empirical/theoretical support.

The paper proposes theoretical findings by extending the existing attention complexity analysis [1] to the VAR model. However, there is a lack of justification for the assumptions (e.g., $d=O(\log n)$, $R=o(\sqrt{\log n})$, $R=\Theta(\sqrt{\log n})$, “Assume each entry in the matrices can be represented using $O(\log n)$ bits.” in Lemma C.7, etc.).

For example, n (the height and width of the last VQ code map) is 16, 32, and 64, depending on the generated image resolution, and d (embedding size of each token, or hidden dimension) varies from 1024 to 1920 as described in Equation (7) of [2]. Under such realistic settings, the assumption employed in this paper, which is $d=O(\log n)$, is not satisfied.

The authors should explain the theoretical findings under the assumptions that match the original paper [2]. Alternatively, the authors need to verify the superiority of VAR models under the assumptions presented in this paper, as the experiments in the original paper [2] – e.g., generating images with $d=O(\log n)$ and showing the performance using various metrics such as FID, IS, or visualizations.

W2. The novelty is unclear.
The contribution of this paper is to propose the computational limit and probably efficient criteria of VAR models [2]. Regarding the computational limit of VAR models, it is not surprising that the reported result, $O(n^{4 - \Omega(1)})$, nearly matches $O(n^4)$ reported in [2]. Fast Attention Computation Theorem (Lemma E.2) used to accelerate the running time of the VAR Transformer to $O(n^{2+o(1)})$ has already been proposed in [1]. The authors should explain what new theoretical contribution was made in deriving the probably efficient criteria. In other words, which parts, such as stage-wise runtime accounting, $\ell_\infty$ error propagation across up-sampling/decoder, are new beyond applying [1]?

W3. The presentation should be improved.
The writing includes undefined notations (e.g., $q$ and $L_k$ in Theorem 4.5), inconsistent notation (e.g., the upsampled output of $X_r$ is denoted as $Y_{r+1}$ in Definition 3.2, but as $Y_r$ in Definition 3.3), confusion in the use of terminology (e.g., VAR transformer and VAR model in Theorem 4.5 and its proof).

[1] Alman, Josh, and Zhao Song. "Fast attention requires bounded entries." Advances in Neural Information Processing Systems 36 (2023): 63117-63135.
[2] Tian, Keyu, et al. "Visual autoregressive modeling: Scalable image generation via next-scale prediction." Advances in neural information processing systems 37 (2024): 84839-84865.

**Questions:**

Please refer to W1, W2, and W3.

---

> ### Author Response · Authors · 2025-12-01
>
> Thank you for your thoughtful feedback. Your comments are very helpful and much appreciated. We will address these in the next version.

---

### Official Review · Reviewer_KNm4 · 2025-11-05

**Soundness:** 2
**Presentation:** 1
**Contribution:** 1
**Rating:** 2
**Confidence:** 3

**Summary:**

This paper provides a fine-grained complexity analysis of Visual Autoregressive (VAR) models. It establishes computational hardness in approximating VAR inference under the Strong Exponential Time Hypothesis (SETH), showing that truly sub-quartic computation is impossible when $d=O (\log n)$ and $R=\theta ( \sqrt {{\log n}} )$. Conversely, under the condition $R=\theta ( \sqrt {{\log n}} )$, the authors demonstrate that near-quadratic-time approximation with bounded additive error is possible via low-rank approximation. This work offers theoretical insights into computational bounds and acceleration criteria for VAR models.

**Strengths:**

- Provides the first formal fine-grained complexity analysis for VAR models.
- Clearly identifies theoretical lower bounds for VAR computation under SETH.
- Shows conditions where VAR can be approximated in $O (n^2+o(1) )$ time, which is a meaningful theoretical contribution.
- Structured and mathematically rigorous, with detailed proofs and model decomposition across VAR stages.

**Weaknesses:**

- The theoretical contribution strongly depends on existing hardness results for attention (Alman & Song), raising concerns about originality beyond applying those results to VAR.
- No experiments or empirical evidence are provided, making it difficult to validate whether the theoretical phase-transition behavior aligns with real VAR systems or whether perceptual quality is preserved.
- The assumption $d=O (\log n)$ does not reflect practical VAR architectures, where token dimension is often decoupled from input resolution and vocabulary size plays a more essential role. The vocabulary size, critical in VQ-based generative models, is ignored despite being central in prior works such as VQ-GAN and RQ-Transformer.
- Inconsistent terminology (Auto-regressive / Autoregressive / AutoRegressive) and minor format issues (bullet points in abstract).
- Related work survey omits foundational works in approximate attention beyond recent papers; diffusion subsection seems tangential and could be shortened.

**Questions:**

- Without relying on Hardness of Approximate Attention (Alman & Song 2023), what is the core novel theoretical contribution specific to VAR?
- How do the results extend to realistic settings where embedding dimension and codebook size do not scale as $O(\log n)$?
- Can the authors provide empirical evidence or even toy experiments confirming that the proposed approximations preserve generation fidelity in VAR?
- How sensitive are the bounds to practical architectural variants of VAR (multi-scale tokenization, vocabulary size, quantization depth)?

---

> ### Author Response · Authors · 2025-12-01
>
> Thank you for your thoughtful feedback. Your comments are very helpful and much appreciated. We will address these in the next version.

---

### Note · Authors · 2025-12-01

I have read and agree with the venue's withdrawal policy on behalf of myself and my co-authors.